# Multi-Modal One-Shot Federated Ensemble Learning for Medical Data with Vision Large Language Model

## Abstract

Federated learning (FL) has attracted considerable interest in the medical domain due to its capacity to facilitate collaborative model training while maintaining data privacy. However, conventional FL methods typically necessitate multiple communication rounds, leading to significant communication overhead and delays, especially in environments with limited bandwidth. One-shot federated learning addresses these issues by conducting model training and aggregation in a single communication round, thereby reducing communication costs while preserving privacy. Among these, one-shot federated ensemble learning combines independently trained client models using ensemble techniques such as voting, further boosting performance in non-IID data scenarios. On the other hand, existing machine learning methods in healthcare predominantly use unimodal data (e.g., medical images or textual reports), which restricts their diagnostic accuracy and comprehensiveness. Therefore, the integration of multi-modal data is proposed to address these shortcomings. Additionally, vision large language models (vLLMs) have emerged as powerful tools due to their ability to interpret and generate textual descriptions from visual data, making them invaluable for creating textual reports from medical images. In this paper, we introduce `FedMME`, an innovative one-shot multi-modal federated ensemble learning framework that utilizes multi-modal data for medical image analysis. Specifically, `FedMME` capitalizes on vision large language models to produce textual reports from medical images, employs a BERT model to extract textual features from these reports, and amalgamates these features with visual features to improve diagnostic accuracy. Experimental results show that our method demonstrated superior performance compared to existing one-shot federated learning methods in healthcare scenarios across four datasets with various data distributions. For instance, it surpasses existing one-shot federated learning approaches by more than 17.5% in accuracy on the RSNA dataset when applying a Dirichlet distribution with ($\alpha = 0.3$).

## 1 Introduction

Federated learning (McMahan et al., 2017) is a paradigm that allows multiple distributed clients to collaboratively train a global model without sharing their local data (Wang et al., 2022; Balkus et al., 2022). This approach encompasses various scenarios such as parallel federated learning (Li et al., 2020), sequential federated learning (Wang et al., 2024a), and federated ensemble learning (Wang et al., 2023). As depicted in Fig. 1, federated ensemble learning involves each participant independently training a model on their respective local datasets. Subsequently, these models are combined on a central server to perform ensemble learning techniques, including voting (Raza, 2019) or stacking (Cui et al., 2021). Federated ensemble learning has gained substantial popularity in privacy-sensitive areas including finance (Gadekallu et al., 2021) and edge computing (Alam et al., 2023).

Recently, federated learning has gained significant attention in the medical field (Pfitzner et al., 2021; Sheller et al., 2020) because it enables the development of more comprehensive and accurate models by integrating data from different medical institutions without compromising data privacy. In contrast, traditional centralized machine learning (Drainakis et al., 2020) requires data aggregation on a central server for training, which presents substantial privacy, legal, and security challenges when handling sensitive medical data. Federated

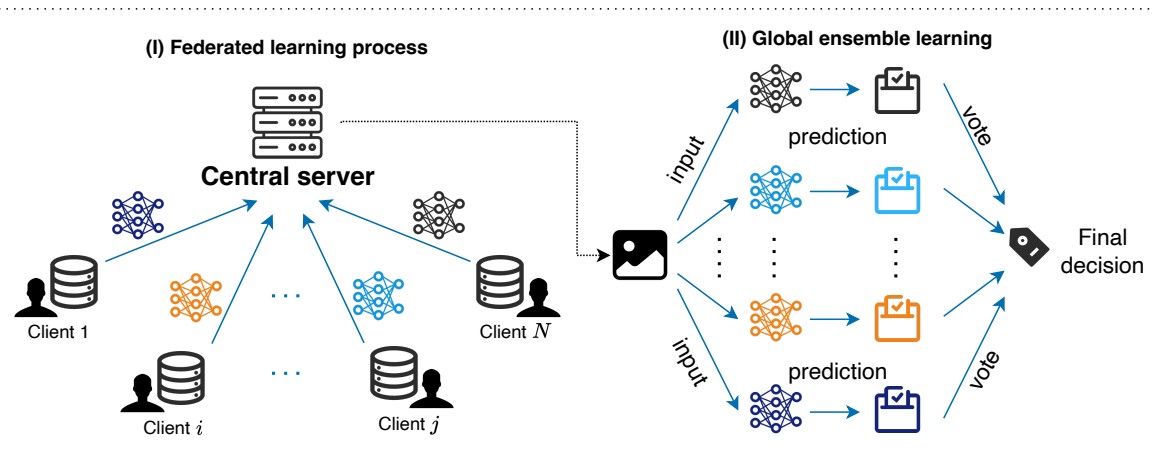

Figure 1: Overview of the one-shot federated ensemble framework. (I) Federated learning process: each client trains a local model using its private dataset and transmits the model to the central server. (II) Global ensemble learning: the server aggregates model outputs from all clients using a voting mechanism to produce the final decision.

learning mitigates these challenges by allowing multiple medical institutions to perform model training in their local environments, exchanging only model updates rather than raw data. This method effectively safeguards data privacy and produces a robust global model or ensemble team that surpasses any model developed by individual institutions alone (Abaoud et al., 2023).

Despite the effectiveness of federated learning in protecting privacy and enabling distributed training for healthcare, it faces challenges related to substantial communication overhead and time delays, especially in environments with limited network bandwidth (Kim et al., 2024). To address the risks of attack and high communication costs associated with traditional federated learning, the concept of one-shot federated learning has been proposed (Guha et al., 2018; Dennis et al., 2021). This communication-efficient method conducts model training and aggregation in a single communication round, significantly reducing overhead and enhancing data privacy (Zhang et al., 2022). Consequently, employing one-shot federated ensemble learning for the development of healthcare applications holds considerable promise.

In the medical field, federated learning methods commonly rely on unimodal data for training, such as solely using medical images (Chakravarty et al., 2021; Ke et al., 2021; Kang et al., 2023) or textual reports (Sui et al., 2020). However, this unimodal approach results in a significant loss of valuable information. For example, analyzing medical images without the context provided by patient history or textual reports can lead to incomplete diagnostics or overlook critical correlations between visual cues and underlying conditions (Reale-Nosei et al., 2024). This information bottleneck restricts a comprehensive understanding and accurate interpretation of a patient's health status.

Consider a scenario in which a CT scan detects an anomaly that initially appears benign. Without incorporating textual data, such as the patient's symptoms or genetic predisposition from their medical history, physicians may overlook the need for further investigation of a potentially serious condition (Rakowski et al., 2006). Similarly, when relying solely on textual reports, subtle indicators in the imagery might be entirely missed (Wu et al., 2024a). By integrating various data types—such as medical images, textual reports, and sensory data from wearable devices—healthcare models can obtain a more comprehensive understanding of patient health. This integrative approach enables the cross-validation and enrichment of datasets (Singh et al., 2020), leading to insights that cannot be achieved through the analysis of unimodal data alone. Therefore, incorporating multi-modal data into one-shot federated learning frameworks for healthcare applications is imperative to achieve more accurate and reliable diagnostic and predictive outcomes.

In recent years, large language models (LLMs) have demonstrated substantial potential in natural language processing (Brown et al., 2020; Chowdhery et al., 2023; Hoffmann et al., 2022) and multi-modal (Hu et al., 2024; Ge et al., 2024) tasks. These models excel in generating high-quality textual reports and in extracting and integrating complex semantic information from both textual and visual data, particularly when augmented with their vision-based counterparts, such as Llama-3.2-11B-Vision (Chi et al., 2024). The advanced capabilities of LLMs offer new opportunities for multi-modal analysis in the medical domain (Ghosh et al., 2024). A medical analysis framework incorporating LLMs can produce detailed imaging reports, thereby enhancing diagnostic accuracy and interoperability (Waldock et al., 2024). Hence, utilizing these advanced vision large language models within medical settings could substantially improve therapeutic effects and streamline clinical workflows.

In this paper, we introduce `FedMME`, an innovative *Fed*erated *M*ulti-*M*odal *E*nsemble learning framework for medical image analysis. Our framework enhances medical data analysis by incorporating both visual and textual features, with the help of vision large language models. Specifically, our method employs these models to generate textual descriptions from medical images, which are then integrated with visual data for model training. This approach significantly enhances prediction accuracy and robustness in medical image analysis. Additionally, our framework effectively harnesses multi-modal information while maintaining low communication costs, making it particularly suited for privacy-sensitive and resource-constrained medical applications such as medical image analysis.

Our main contributions can be summarized as follows:

- We introduce `FedMME`, a novel one-shot, multi-modal federated learning framework designed for medical image analysis. To the best of our knowledge, this is the first work to systematically explore the one-shot federated ensemble learning for training multi-modal models.
- We innovatively employ large vision language models to generate more accurate medical reports from images, thereby extracting superior textual features that enhance overall performance.
- We conduct comprehensive experiments across four datasets with various data distributions. Our method demonstrates superior performance compared to existing one-shot FL methods in healthcare scenarios.

## 2 Related Work

Federated Learning (FL) (McMahan et al., 2017) is a decentralized machine learning framework that aims to train models without centralizing user data, thus preserving privacy and reducing communication costs. In recent years, FL has made significant advances in addressing data heterogeneity (Li et al., 2020; Karimireddy et al., 2020; Ye et al., 2023), enhancing privacy protection (Geyer et al., 2017; Wei et al., 2020), and optimizing communication efficiency (Caldas et al., 2018). Despite these advancements, traditional FL often necessitates multiple rounds of communication to achieve global optimization, presenting challenges in communication-limited environments. To overcome this, one-shot federated learning has been introduced (Guha et al., 2019; Su et al., 2023), which accomplishes model aggregation in a single round of global communication, thereby significantly reducing communication overhead. Specifically, FedDISC (Yang et al., 2024) investigates one-shot semi-supervised federated learning using a pre-trained diffusion model. Additionally, DENSE (Zhang et al., 2022) presents a data-free one-shot federated learning approach through knowledge distillation. FedISCA (Kang et al., 2023) is the state-of-the-art one-shot federated learning framework for medical applications, which effectively addresses data heterogeneity by generating synthetic data and adapting client models with the help of knowledge distillation. All these methods support the rapid deployment of federated models in scenarios where frequent communication or data sharing is impractical or undesirable, often due to privacy considerations or bandwidth constraints.

Ensemble learning seeks to combine multiple weak base models to create a more robust model. This approach has been extensively researched for decades and is applicable in various contexts, including federated learning. Traditional ensemble learning techniques include Voting (Raza, 2019), Bagging (Breiman, 1996), Boosting (Schapire, 2013), and Stacking (Wolpert, 1992). Federated Ensemble Learning (Wang et al., 2023) extends these traditional techniques to the decentralized setting of federated learning. FedDF (Lin et al., 2020) introduces ensemble distillation, a method for robustly fusing heterogeneous client models in federated learning by training a central model on unlabeled data using client model predictions. FedEL (Wu et al.,

2024b) introduces a federated ensemble learning approach that trains diverse weak learners across non-IID client data and combines them into a robust global model to improve performance under data heterogeneity. In this study, we explore the training schemes of multi-modal models in federated ensemble learning.

Large Language Models (LLMs) (Brown et al., 2020; Chowdhery et al., 2023; Ouyang et al., 2022) have garnered recognition for their advanced capabilities in Natural Language Processing (NLP) tasks. Their widespread adoption is primarily due to their proficiency in generating coherent text, comprehending complex linguistic structures, and providing contextually relevant responses. Numerous techniques have been developed to enhance the generative performance of LLMs and broaden their application areas. For instance, Chain-of-Thought (Wei et al., 2022) illustrates LLMs' ability to formulate a distinct *"thought process"* to tackle problems. WebGPT (Nakano et al., 2021) employs LLMs to interact with web browsers, navigate web pages, and address complex queries effectively. Beyond conventional text generation tasks, Vision Large Language Models (Wang et al., 2024b; Chi et al., 2024) extend the capabilities of traditional LLMs by incorporating visual comprehension, thus facilitating the convergence between textual and visual modalities. Minigpt-4 (Zhu et al., 2023) integrates a vision encoder with an LLM to augment visual understanding and multi-modal generation. Furthermore, Video-chatgpt (Maaz et al., 2023) introduces a model that merges video-adapted vision encoders with LLMs for intricate video analysis and dialogue generation. These models have significant utility in assisting with human tasks, including those in the medical field.

## 3 Methodology

### 3.1 Problem Definition

As shown in Fig. 1, in the one-shot federated ensemble learning setting, there are $N$ distributed clients (or parties) and a central model server, each client has its private dataset $D_i = \{(x_k, y_k)\}_{k=1}^{n_i}$, where $n_i$ represents the size of the dataset for client $i$. The objective is to construct an optimal ensemble team $\mathcal{M} = \{M_i\}_{i=1}^{N}$, on dataset $\mathcal{D} = \{D_i\}_{i=1}^{N}$. This ensemble team comprises $N$ models, each independently trained by individual clients. In this setting, it is crucial to maintain all client data on their respective local devices, while only the models trained locally are sent to the server. In the one-shot setting, every client is allowed to communicate with the server only once. The objective is achieved by minimizing the loss of the ensemble team $\mathcal{M}$ on dataset $\mathcal{D}$, defined as follows:

$$\min \mathbb{E}_{(x,y)\sim\mathcal{D}} \ell(f_{\mathcal{M}}(x), y), \tag{1}$$

where $\ell$ is the loss function, $f_{\mathcal{M}}(x)$ denotes the prediction function based on $\mathcal{M}$, obtained as the result of ensemble learning, such as voting.

### 3.2 Proposed Method: FedMME

Numerous studies (Huang et al., 2024; Koga, 2025; Zeiser, 2024) suggest that the integration of textual reports with image features significantly improves the performance of image classification compared to employing image features alone. Consequently, it is crucial to utilize multi-modal data when developing diagnostic tools in the healthcare industry. As depicted in Fig. 2, we present a one-shot multi-modal federated ensemble framework, FedMME, to address the issue outlined in Sec. 3.1. In this framework, each client independently trains a multi-modal model on its local dataset utilizing a vision large language model, subsequently sending the trained model to a centralized server. On the server side, these models are aggregated to form an ensemble team. The ultimate decision is then determined through a voting mechanism involving the models within this ensemble team.

As shown in Fig. 2, our method encompasses two primary phases: (1) *Feature Extraction*: this phase involves extracting two types of features. Initially, high-dimensional visual features are extracted from medical images with conventional vision models. Subsequently, a vision large language model is employed to generate a report for the image, from which textual features are derived. (2) *Feature Fusion*: during this phase, visual and textual features are concatenated to create a comprehensive feature layer used for multi-modal classification. To ensure that visual features maintain a dominant role in the prediction process—as the textual modality

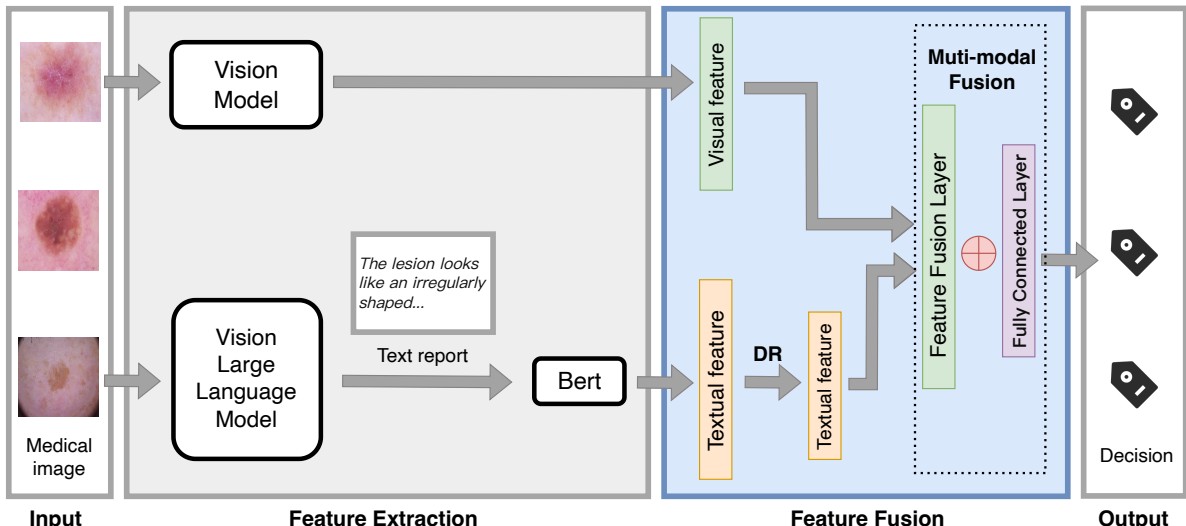

Figure 2: Overview of our proposed framework, `FedMME`, which is structured into two principal phases: feature extraction and feature fusion. In the feature extraction phase, visual features are obtained from images using conventional vision models, such as ResNet-18, and accompanying textual reports are produced via a sophisticated vision large language model. Textual features are subsequently extracted from these reports using a model designed for textual feature extraction, such as BERT. During the feature fusion phase, these visual features are combined with dimensionality-reduced textual features to create a cohesive feature representation. This combined representation is then processed through a fully connected layer, which enables classification.

---

**Algorithm 1:** One-shot multi-modal federated ensemble framework

---

**Input:** Datasets $\mathcal{D} = \{D_i\}_{i=1}^N$ with $D_i = \{(x_k, y_k)\}_{k=1}^{n_i}$, learning rate $\eta$, number of local iterations $E_{local}$, batch size $B$, number of classes $Y$

**Output:** The ensemble team $\mathcal{M} = \{M_i\}_{i=1}^N$

**On server side:**
**for** *client $i = 1 : N$ in parallel* **do**
   |   $M_i \leftarrow$ **LocalTraining**$(D_i)$
**end**
// The inference process for dataset $D = \{x_k\}_{k=1}^n$
**for** $k = 1 : n$ **do**
   |   Initialize $v = \{0\}_{y=1}^Y$
   |   **for** $i = 1 : N$ **do**
   |     |   $y_i = M_i(x_k)$ // refer to Algorithm 2
   |     |   $v[y_i] \leftarrow v[y_i] + 1$
   |   **end**
   |   $y_k = \arg\max_y v[y]$
**end**

**On client side:**
**LocalTraining**$(D_i)$:
Initialize $M_i$
**for** *each local epoch $t$ from* $1$ **to** $E_{local}$ **do**
   |   Split $D_i$ into $\lceil n_i/B \rceil$ batches: $\{B_1, B_2, \ldots, B_s\}$
   |   **for** *batch $B_j$ in* $\{B_1, B_2, \ldots, B_s\}$ **do**
   |     |   $B_j = \{(x_k, y_k)\}_{k=1}^{|B_j|}$
   |     |   $\hat{y}_k = M_i(x_k)$ for all $x_k \in B_j$
   |     |   $L(B_j) = \frac{1}{|B_j|} \sum_{y_k \in B_j} \ell(\hat{y}_k, y_k)$
   |     |   $M_i \leftarrow M_i - \eta \nabla L(B_j)$
   |   **end**
**end**
return $M_i$ to the server

---

is intended to serve as an auxiliary—we perform dimensionality reduction on the textual features before integration, thus minimizing their impact. Finally, a fully connected layer is attached to this comprehensive feature layer to produce the final prediction results.

---

**Algorithm 2:** `FedMME`

---

**Input:** Original image $x$
**Output:** The prediction result $y_{pred}$
1. Extract the image features with a traditional vision model
$f_{visual} = VisionModel(x)$
2. Extract the textual features with a vision large language model
$r = VisionLLM(x)$ // r: the report of images
$f_{textual} = BERT(r)$
3. Integrate the extracted visual and textual features
$f^*_{textual} = DR(f_{textual})$// $DR(\cdot)$: Dimensionality Reduction
$f_{combine} = f_{visual} + f^*_{textual}$
4. Get the prediction result
$y_{pred} = FC(f_{combine})$// $FC(\cdot)$: fully connected layer
return $y_{pred}$

---

### 3.2.1 Feature Extraction

***Visual Feature Extraction***: We utilize a conventional vision model, such as the ResNet-18 (He et al., 2016) model with the final layer removed, to extract visual features from medical images. This approach is extensively employed in image processing tasks. For a given medical image $x$, the extracted visual features can be represented as:

$$f_{visual} = VisionModel(x), \tag{2}$$

where $f_{visual} \in \mathbb{R}^{d_1}$ denotes the feature vector extracted by the vision model, and $d_1$ is the dimension of the feature vector. The vision model is pre-trained to accurately capture critical features in medical images, thereby ensuring optimal performance in subsequent tasks. This model will also be incorporated into the subsequent multi-modal training process to further enhance the effectiveness of our ensemble team.

***Text Report Generation***: Our method seeks to enhance the diagnostic accuracy of medical images by incorporating textual features, and vision large language models have been demonstrated to significantly enhance feature richness by interpreting the context within images (Zhou, 2024; Ding et al., 2024). These models function by comprehending and generating textual descriptions from visual data, thereby bridging the gap between visual perception and linguistic comprehension. By leveraging these models, our system can identify subtle details often overlooked by traditional image-only methods. However, employing solely a vision large language model for the classification of medical images generally results in suboptimal outcomes. For example, when evaluated under the identical experimental conditions described in our study when the Dirichlet distribution $\alpha$ =0.6, the Llama-3.2-8B-Vision-Instruct (Chi et al., 2024) model only achieves a classification accuracy of 29% on the Blood (Yang et al., 2023) dataset. This performance is significantly inferior to the approximate 80% accuracy achieved by purely visual classification models such as ResNet-18. Consequently, it is inadvisable to rely exclusively on vision large language models for image classification tasks. Instead, these models should be incorporated as ancillary training resources within a multi-modal framework to augment the efficacy of vision models. Therefore, after the extraction of image features, we utilize a vision large language model, such as the Llama-3.2-11B-Vision-Instruct model, to generate textual reports for the medical image:

$$r = VisionLLM(x), \tag{3}$$

where $r$ denotes the generated report for the medical image $x$.

***Textual Feature Extraction***: The text generated by the large vision language model is presented in textual format rather than as visual features like those extracted by the vision model. Consequently, we cannot directly merge it with the visual features for input into a multi-modal model. To address this, we

utilize a pre-trained BERT (Devlin, 2018; Onan, 2023) model to derive textual features from the report. BERT has exhibited an exceptional ability to grasp the context and semantics of text across a multitude of natural language processing applications. The textual features produced by BERT are in the form of embeddings—dense representations that encapsulate various dimensions and interrelationships within the textual data. These embeddings can then be concatenated with the visual features for prediction purposes. The process of extracting textual features can be depicted as follows:

$$f_{textual} = BERT(r), \tag{4}$$

where $f_{textual} \in \mathbb{R}^{d_2}$ denotes the feature vector extracted by BERT, $r$ is the textual description generated by the vision large language model. $d_2$ represents the dimension of the textual feature vector.

### 3.2.2 Feature Fusion

***Dimensionality Reduction***: When merging textual and visual features during training, it is crucial to ensure that the textual elements do not overpower or displace visual information as the main modality. If textual features dominate, it could significantly compromise the generalization capability of the model (Rahman et al., 2020; Fei et al., 2022). For instance, in employing the ResNet-18 model for visual features extraction and the BERT model for textual features, visual features from ResNet are noted to have a dimensionality of 512, in contrast to the 768 of BERT. This greater dimensionality of textual features might inadvertently suppress the visual modality, negating our original purpose.

Therefore, to prevent textual features from overshadowing visual features, we apply a dimension-reduction strategy on the textual features. This reduction aims to curtail their influence on the overall effectiveness of the model, thereby ensuring that visual features maintain a more prominent role in driving the model's predictions. The dimensionality reduction process is outlined as follows:

$$f^*_{textual} = DR(f_{textual}), \tag{5}$$

where $f^*_{textual}$ represents the textual features after the dimensionality reduction process, denoted as $DR(\cdot)$.

***Feature Concatenation***: After reducing the dimensionality of textual features, we concatenate them with visual features to form a unified feature representation for training our multi-modal model. This integration harnesses the strengths of each modality, thereby enhancing both the robustness and accuracy of the model predictions. The combined feature vector is represented as follows:

$$f_{combine} = f_{visual} + f^*_{textual}, \tag{6}$$

where $f_{combine}$ denotes the new feature vector created by concatenating the visual features with the dimensionality-reduced textual features.

Finally, we attach a fully connected layer to the integrated feature vector to produce the final prediction result for the corresponding image.

$$y_{pred} = FC(f_{combine}), \tag{7}$$

where $FC(\cdot)$ represents the application of a fully connected layer to the concatenated features.

Alg. 1 presents a comprehensive overview of the federated ensemble learning framework and the inference mechanisms using our multi-modal model, where $M_i$ denotes our designed multi-modal model on client $i$. Meanwhile, Alg. 2 details the procedure of our `FedMME` framework for generating and handling multi-modal data features. After obtaining the entire ensemble team $\mathcal{M}$, ensemble learning techniques can be applied to derive the final prediction outcomes, for example, through the utilization of voting methods.

# 4 Experiments

## 4.1 Experiments Setup

**Datasets.** We selected four representative medical datasets to evaluate our framework: Blood and Derma from the MedMNIST dataset (Yang et al., 2023), RSNA (Rsna, 2019), and Diabetic (Dugas et al., 2015). These datasets cover a range of medical application scenarios. To closely mimic real-world conditions, we conducted extensive experiments across various Non-IID settings—a prevalent challenge in federated learning—to comprehensively evaluate the efficacy of our method. Specifically, we employed a Dirichlet distribution to allocate datasets among $N = 5$ clients using concentration parameters $\alpha = 0.6, 0.3$, and $0.1$. This approach enabled us to explore different levels of data heterogeneity and rigorously test the robustness of our methodology.

**Settings.** In our experiments, the vision model employs the ResNet-18 (He et al., 2016) architecture to extract visual features from original medical images. Local client models were trained for 100 epochs with SGD optimizer using learning rate (LR) 1e-3 and batch size 128. Additionally, we utilize the Llama-3.2-11B-Vision-Instruct (Chi et al., 2024) model to generate textual reports from medical images and employ a BERT (Devlin, 2018) model to extract textual features from these reports. Finally, we utilize the equal-weight voting (Raza, 2019) strategy to implement the ultimate ensemble learning process.

All our experiments are running on a single machine with 1TB RAM and 256 cores AMD EPYC 7742 64-Core Processor @ 3.4GHz CPU. The GPU we used is NVIDIA A100 SXM4 with 40GB memory. The environment settings are: Python 3.9.12, PyTorch 1.12.1 with CUDA 11.6 on Ubuntu 20.04.4 LTS.

All the experimental results are the average over three trials.

## 4.2 Baselines

We compared our proposed method with four established baseline methods: **FedAvg** (McMahan et al., 2017), a well-known federated learning algorithm that conducts multi-round optimization of a distributed model by performing local training on clients and using weighted averaging for aggregation on the server. For consistency with our method, we adapted FedAvg to a *one-shot* setting. **DENSE** (Zhang et al., 2022) represents a one-shot federated learning framework relying on a central server to disseminate a global model through knowledge distillation. **DAFL** (Chen et al., 2019) is another one-shot federated learning strategy, in which a teacher model is built from multiple client models using knowledge distillation to develop a global model. **FedISCA** (Kang et al., 2023), the state-of-the-art one-shot federated learning framework, is specifically designed for medical applications and addresses data heterogeneity through synthetic data generation and client model adaptation with the help of knowledge distillation. Additionally, we incorporated **FedEnsemble** into our comparison. This method entails each local client employing a single modal vision model, such as Resnet-18 in our experiments, to facilitate ensemble learning on a central server.

## 4.3 Performance Analysis

Table 1 presents the accuracy results for various datasets across different data partitions and methods. As we can see, our method consistently outperforms all other baselines across all data partitions and datasets. Notably, it surpasses existing one-shot federated learning approaches by more than 17.5% in accuracy on the RSNA dataset when applying a Dirichlet distribution with ($\alpha = 0.3$). This significant improvement underscores the efficacy of our framework in one-shot federated ensemble learning within medical contexts. Furthermore, although the accuracy of all methods tends to decrease as the degree of non-IID increases, our method persistently exhibits superior performance, thereby illustrating its robustness.

It is essential to acknowledge that numerous baseline methods encounter difficulties in achieving high performance on the Diabetic dataset, which stands as the largest dataset embodying real-world data. For instance, some approaches, such as DAFL, only manage to reach accuracy levels akin to that of random guessing, thereby underscoring their inadequacies in handling real-world scenarios. In contrast, our proposed method exhibits strong and consistent performance, significantly outperforming other techniques. More precisely,

Table 1: Test accuracy comparison for different datasets on various data partitions and methods. **Bold** indicates the best accuracy among one-shot FL methods.

| | Dirichlet ($\alpha$=0.6) | | | | Dirichlet ($\alpha$=0.3) | | | | Dirichlet ($\alpha$=0.1) | | | |
|---|---|---|---|---|---|---|---|---|---|---|---|---|
| Dataset | Blood | Derma | RSNA | Diabetic | Blood | Derma | RSNA | Diabetic | Blood | Derma | RSNA | Diabetic |
| FedAvg | 19.48 | 69.23 | 70.39 | 20.01 | 30.51 | 11.02 | 69.31 | 19.99 | 19.81 | 66.88 | 68.55 | 19.98 |
| DAFL | 17.13 | 63.33 | 50.44 | 20.03 | 16.03 | 13.64 | 48.88 | 20.04 | 15.11 | 12.58 | 46.69 | 20.01 |
| DENSE | 34.52 | 64.78 | 55.04 | 23.31 | 30.17 | 12.78 | 51.08 | 23.22 | 28.87 | 11.37 | 47.79 | 21.32 |
| FedISCA | 53.61 | 53.86 | 70.59 | 26.98 | 48.93 | 16.11 | 70.39 | 28.79 | 53.61 | 53.86 | 69.11 | 20.97 |
| FedEnsemble | 84.33 | 67.23 | 83.46 | 24.85 | 71.03 | 66.13 | 70.42 | 25.48 | 54.92 | 67.13 | 68.76 | 20.54 |
| **FedMME** | **87.72** | **71.27** | **85.49** | **29.81** | **80.12** | **71.27** | **87.93** | **31.93** | **68.11** | **69.13** | **71.38** | **21.55** |

Table 2: Performance comparison on the Blood dataset with the different number of clients, where Dirichlet parameter $\alpha = 0.3$.

| Number of Users | FedAvg | DAFL | DENSE | FedISCA | FedEnsemble | FedMME |
|---|---|---|---|---|---|---|
| 5 | 30.51 | 16.03 | 30.17 | 48.93 | 71.03 | **80.12** |
| 10 | 20.22 | 17.11 | 31.28 | 49.99 | 77.17 | **82.72** |
| 20 | 20.11 | 18.29 | 33.25 | 52.47 | 78.16 | **86.23** |

when the Dirichlet parameter is set to $\alpha = 0.3$, our method attains an accuracy of 31.93%, which not only surpasses DAFL by 11% but also exceeds the second-best method, FedISCA, by over 3%.

Meanwhile, we conducted experiments with varying numbers of clients to assess the scalability of our method. Table 2 demonstrates that our approach consistently outperforms other baselines across all scenarios involving different numbers of clients on the Blood dataset under a Dirichlet distribution ($\alpha = 0.3$). This underscores its feasibility for large-scale federated learning environments and attests to its robustness. Additionally, we observed an improvement in the performance of most methods as the number of models increased with more clients. This enhancement can be attributed to the increased diversity within the group of models, which, in turn, leads to superior ensemble or aggregation performance (Wu et al., 2021; Wang et al., 2023), particularly compared to scenarios with fewer clients.

### 4.4 Ablation Studies

#### 4.4.1 Comparative analysis of Vision Large Language Model

In the process of generating textual reports from medical images, we hypothesized that diverse vision large language models might output varying reports when interpreting the same image, potentially impacting the efficacy of our multi-modal training approach. Consequently, we assessed the performance of two vision large language models, Llama-3.2-11B-Vision-Instruct and ChatGPT-4o-Vision, across multiple datasets characterized by a Dirichlet distribution with ($\alpha$ =0.3). Fig. 3 shows that the Llama-3.2-11B-Vision-Instruct model consistently outperformed the ChatGPT-4o-Vision model in all four datasets. This is because ChatGPT-4o-Vision often refrained from responding to domain-specific queries, typically responding with, *"I'm sorry, I can't assist with identifying the type of blood cell from this image"*. However, when we instructed the ChatGPT-4o-Vision model to focus on describing the features of the images, rather than directly categorizing them through tailored prompts, both models achieved comparable performance. This underscores the adaptability of our method across different types of vision large language models.

#### 4.4.2 Effects of different textual feature size

To assess the impact of dimension-reduced textual features on our framework, we conducted experiments in which we reduced the original textual features to various dimensions. Fig. 4 illustrates that when the textual feature size remains below 512, the impact on performance is negligible, with optimal results at a dimension of 128. This indicates that our method maintains robustness across different sizes of textual features. However, performance deteriorates markedly as the textual feature size increases to 512—the point at which it equals the visual feature size. This decline is attributed to the excessive size of the textual features, which nearly

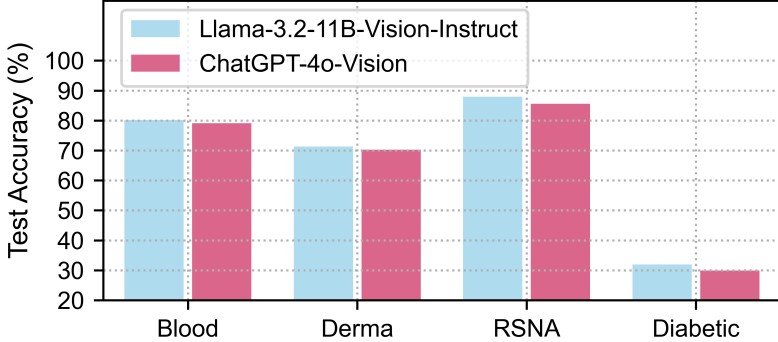

Figure 3: Test Accuracy comparison of `FedMME` with different vision large language models, where Dirichlet parameter $\alpha = 0.3$.

supplants the primary visual modality and consequently degrades overall performance. Thus, maintaining the dimensionality reduction component is crucial for the efficacy of the multi-modal model.

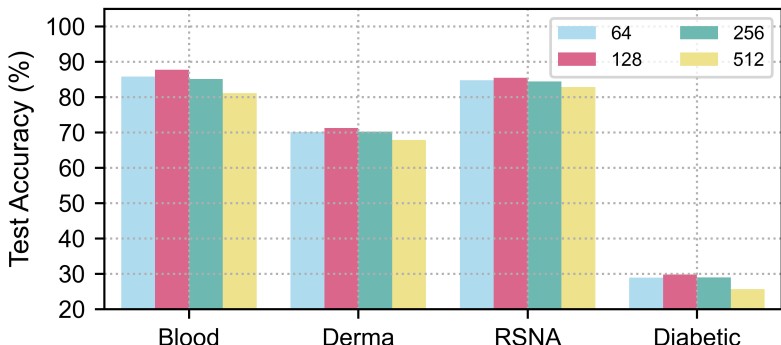

Figure 4: Effect of various textual feature sizes across different datasets, where Dirichlet parameter $\alpha = 0.6$.

### 4.4.3   Training convergence analysis

Fig. 5 illustrates the performance of our method across varying numbers of local training epochs for all evaluated datasets. Notably, our model demonstrates substantial accuracy improvements after just 10 rounds of local training, indicating rapid convergence and sustained high performance with fewer iterations. Additionally, the results underscore the method's robustness, as it maintains consistent performance regardless of changes in the number of local training epochs. This consistency highlights the method's adaptability and reliability in diverse scenarios.

### 4.5   Case Study

In this section, we present a case study to illustrate the effectiveness of the vision large language model incorporated into our framework. Fig. 6 is derived from the Diabetic Retinopathy dataset, primarily utilized to evaluate the severity of Diabetic Retinopathy (DR), a common complication of diabetes. Accurate assessment of DR severity is crucial for devising effective treatment plans for patients.

Fig. 7 and Fig. 8 illustrate the prompt and response utilized by the vision large language model, Llama-3.2-11B-Vision-Instruct, to categorize and analyze the medical image depicted in Fig. 6. In this case, despite the Llama-3.2-11B-Vision-Instruct model incorrectly classifying the severity as *"Moderate diabetic retinopathy"* rather than the correct severity *"Proliferative diabetic retinopathy"*, the text report still provides valuable

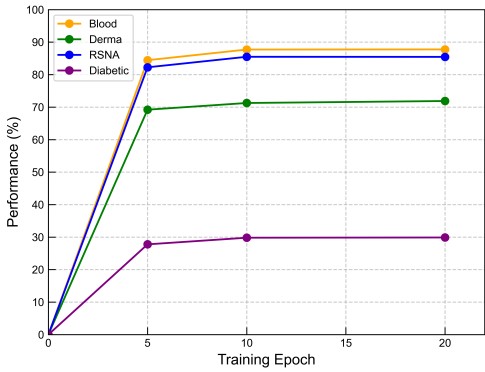

Figure 5: Training convergence analysis by various training epochs, where Dirichlet parameter $\alpha = 0.6$.

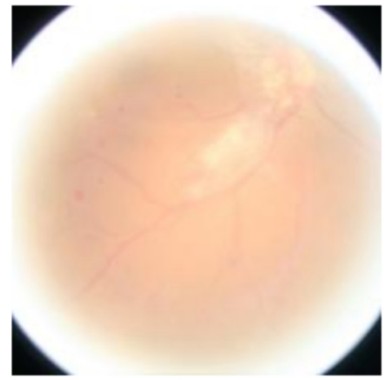

Figure 6: Retinal image with several features that are typical of diabetic retinopathy (DR).

---

**Prompt**

You are a ophthalmologist in a hospital, this image is a retinal image from a possible patient, please review the image to rate the severity of diabetic retinopathy (DR) of this image.

Please use the following format for your diagnosis:

**SEVERITY**: Select from one of the following severity: '0: No diabetic retinopathy', '1: Mild diabetic retinopathy', '2: Moderate diabetic retinopathy', '3: Severe diabetic retinopathy', '4: Proliferative diabetic retinopathy'.
**DIAGNOSIS**: Give a brief diagnosis based on the condition of the retinal image, explain why you think it is of this severity.

---

Figure 7: The prompt used for the Llama-3.2-11B-Vision-Instruct model to recognize the type of the medical image shown in Fig. 6.

insights. It successfully identifies specific retinal features, including hemorrhages and microaneurysms. These detailed descriptions of the images significantly contribute to improving the accuracy of predictions made by multi-modal models. Therefore, our multi-modal training framework, augmented by the vision large language model, demonstrates to be a significant tool in the realm of medical diagnostic applications.

## 5 Conclusion

In this paper, we present a novel multi-modal one-shot federated ensemble learning framework for medical image analysis. By incorporating both visual and textual features with the help of large vision language models, our method not only addresses the challenges posed by data privacy but also improves the robustness and accuracy of medical diagnostics. We demonstrated the effectiveness of our framework across multiple datasets, showing its superior performance in analyzing diverse medical images compared to traditional federated learning methods. To the best of our knowledge, this is the first work to systematically explore the one-shot federated ensemble learning for training multi-modal models.

---

**Answer**

**Step 1: Analyze the retinal image**
The image shows a retinal image with some features that are typical of diabetic retinopathy (DR). There are some microaneurysms and retinal hemorrhages visible in the image, which are common signs of DR. However, there are no signs of neovascularization or vitreous hemorrhage, which are typically seen in more severe cases of DR.

**Step 2: Determine the severity of DR**
Based on the features visible in the image, I would rate the severity of DR as follows:
**SEVERITY:** 2: Moderate diabetic retinopathy.
**DIAGNOSIS:** The patient has moderate diabetic retinopathy. The presence of microaneurysms and retinal hemorrhages indicates that the patient has some level of DR, but the lack of neovascularization or vitreous hemorrhage suggests that the disease is not yet severe. Therefore, I would rate the severity as moderate.

---

Figure 8: The output generated by the Llama-3.2-11B-Vision-Instruct model in response to the prompt presented in Fig. 7 and the medical image depicted in Fig. 6.

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
