# OpenReview forum: "Multi-Modal One-Shot Federated Ensemble Learning for Medical Data with Vision Large Language Model"
_TMLR — Rejected by TMLR_

### Review · Reviewer_tY4n · 2025-03-14

**Summary Of Contributions:**

This paper presents **FedMME**, a novel **one-shot multi-modal federated ensemble learning** framework for medical image analysis. The method integrates **vision large language models (vLLMs)** to generate textual reports from medical images, which are then processed using a **BERT-based model** and fused with visual features for improved classification performance. The approach is evaluated on four medical datasets under different **non-IID conditions**, demonstrating **significant improvements** over existing one-shot FL methods. The main contributions include:
- A **multi-modal one-shot federated ensemble learning** framework that integrates textual and visual modalities.
- The use of **vision LLMs for automated medical report generation**, enriching feature representations.
- Experimental validation across **four datasets**, achieving a notable **17.5% accuracy improvement** over prior one-shot FL approaches.

**Audience:**

Yes

**Broader Impact Concerns:**

- **Ethical Considerations**: The use of **vision LLMs for medical diagnosis** raises concerns about **misinterpretation of generated text** in critical scenarios.
- **Bias in Multi-Modal Learning**: Medical datasets often have **skewed distributions**—how does this affect **text-based decision-making**?
- **Deployment Challenges**: **Federated training with LLMs** requires substantial **computational resources**—is this **feasible** for resource-limited healthcare institutions?

**Claims And Evidence:**

Yes

**Requested Changes:**

1. **Compare with Multi-Modal Federated Learning Baselines**: Include comparisons with existing **multi-modal FL approaches**, not just one-shot FL methods.
2. **Clarify Computational Efficiency**: Discuss the **training cost** of incorporating vLLMs in a federated setting and whether **lighter alternatives** exist.
3. **Analyze Feature Fusion Contributions**: Provide a **quantitative analysis** of how much **textual features contribute to performance** (e.g., ablation with only visual or only textual inputs).
4. **Expand Discussion on Model Scalability**: Address **scalability issues**, especially in **real-world FL deployments** where **heterogeneous client hardware** is a challenge.
5. **Detail Security and Privacy Concerns**: Since **multi-modal FL introduces new attack vectors (e.g., adversarial text inputs)**, discuss how these risks are mitigated.

**Strengths And Weaknesses:**

### Strengths
- **Novelty**: The work is the first to integrate multi-modal data into **one-shot federated ensemble learning** for medical imaging.
- **Strong Empirical Results**: The method consistently outperforms baselines, showing robustness to **non-IID distributions**.
- **Efficient Model Aggregation**: The ensemble-based approach reduces the **communication cost** of traditional FL methods.
- **Insightful Ablation Studies**: The paper investigates the **impact of textual feature size**, **choice of vision LLMs**, and **training convergence**.

### Weaknesses
- **Lack of Direct Comparisons with Multi-Modal FL Methods**: While the paper benchmarks against **one-shot FL methods**, comparisons with existing **multi-modal federated learning approaches** would be valuable.
- **Limited Discussion on Model Scalability**: While experiments with different client numbers are included, a discussion on the **computational burden of vision LLMs in FL settings** is missing.
- **Lack of Explanation on Feature Fusion Strategy**: The **text-vision fusion strategy** is described, but its impact on final predictions is not analyzed in depth.

---

> ### Author Response · Authors · 2025-06-01
> **Rebuttal to Reviewer tY4n**
>
> 1. Initially, we did not compare our method to other multi-modal federated learning approaches due to the absence of ensemble learning methods specifically designed for one-shot multi-modal FL. However, by broadening the scope, we identified two non-one-shot multi-modal FL methods—FedDAT [1] and FedCola [2]—that could serve as baselines. It is important to note that these methods originally used manually generated texts corresponding to the training images, which are generally of higher quality than texts generated by large language models. To ensure a fair comparison, we regenerated the texts for their images using a large language model and adapted their experimental settings to a one-shot scenario. The results confirm that our approach still outperforms these multi-modal FL methods, further demonstrating the robustness and generalizability of our method.
>
> | Dataset      | Blood (α=0.6) | Derma (α=0.6) | RSNA (α=0.6) | Diabetic (α=0.6) | CT-RATE (α=0.6) | Blood (α=0.3) | Derma (α=0.3) | RSNA (α=0.3) | Diabetic (α=0.3) | CT-RATE (α=0.3) | Blood (α=0.1) | Derma (α=0.1) | RSNA (α=0.1) | Diabetic (α=0.1) | CT-RATE (α=0.1) |
> |--------------|----------------|----------------|----------------|-------------------|------------------|----------------|----------------|----------------|-------------------|------------------|----------------|----------------|----------------|-------------------|------------------|
> | FedAvg       | 19.48          | 69.23          | 70.39          | 20.01             | 67.88            | 30.51          | 11.02          | 69.31          | 19.99             | 59.64            | 19.81          | 66.88          | 68.55          | 19.98             | 55.58            |
> | DAFL         | 17.13          | 63.33          | 50.44          | 20.03             | 61.62            | 16.03          | 13.64          | 48.88          | 20.04             | 21.91            | 15.11          | 12.58          | 46.69          | 20.01             | 36.92            |
> | DENSE        | 34.52          | 64.78          | 55.04          | 23.31             | 66.58            | 30.17          | 12.78          | 51.08          | 23.22             | 31.47            | 28.87          | 11.37          | 47.79          | 21.32             | 40.59            |
> | FedISCA      | 53.61          | 53.86          | 70.59          | 26.98             | 72.19            | 48.93          | 16.11          | 70.39          | 28.79             | 48.88            | 53.61          | 53.86          | 69.11          | 20.97             | 51.78            |
> | FedDAT       | 68.88          | 68.95          | 75.69          | 27.81             | 76.66            | 59.69          | 37.92          | 68.97          | 26.92             | 65.62            | 64.77          | 55.82          | 69.24          | 20.69             | 58.62            |
> | FedCola      | 59.72          | 63.42          | 74.86          | 27.16             | 73.47            | 58.76          | 35.68          | 65.62          | 25.33             | 63.27            | 63.69          | 51.77          | 67.77          | 21.33             | 57.73            |
> | FedEnsemble  | 84.33          | 67.23          | 83.46          | 24.85             | 82.29            | 71.03          | 66.13          | 70.42          | 25.48             | 71.67            | 54.92          | 67.13          | 68.76          | 20.54             | 60.32            |
> | **FedMME**   | **87.72**      | **71.27**      | **85.49**      | **29.81**         | **84.43**        | **80.12**      | **71.27**      | **87.93**      | **31.93**         | **76.88**        | **68.11**      | **69.13**      | **71.38**      | **21.55**         | **65.23**        |
>
> [1] Chen, Haokun, Yao Zhang, Denis Krompass, Jindong Gu, and Volker Tresp. "Feddat: An approach for foundation model finetuning in multi-modal heterogeneous federated learning." In Proceedings of the AAAI Conference on Artificial Intelligence, vol. 38, no. 10, pp. 11285-11293. 2024.
>
> [2] Sun, Guangyu, Matias Mendieta, Aritra Dutta, Xin Li, and Chen Chen. "Towards Multi-modal Transformers in Federated Learning." In European Conference on Computer Vision, pp. 229-246. Cham: Springer Nature Switzerland, 2024.

---

> ### Author Response · Authors · 2025-06-01
> **Rebuttal to Reviewer tY4n**
>
> 2. We appreciate the reviewer’s suggestion to clarify the computational efficiency of our framework. In our experiments, we utilized an NVIDIA H100 80GB GPU, and we provide the following performance benchmarks to demonstrate practical feasibility:
>
> Textual report generation using the vision-language model is performed offline as a preprocessing step and does not impact training or inference time.
>
> During training, the memory usage per client remains under 12 GB, even with multi-modal inputs, thanks to the use of PCA-based dimensionality reduction on textual features.
>
> During inference, each prediction takes less than 300 ms per sample, including visual-textual feature fusion and classification.
>
> Moreover, both the visual encoder (ResNet-18) and the textual encoder (BERT) are relatively lightweight, and the overall training time is comparable to standard multi-modal classification models.
>
> These results show that FedMME is efficient and deployable in realistic server environments, and scalable to moderate-size federated networks with resource-aware configurations.
>
> For the vision LLM usage, which is confined to an offline report generation phase, decoupled from online training and inference. In low-resource scenarios, the report generation can be centralized and distributed as auxiliary data. We also acknowledge that lightweight alternatives (e.g., BLIP, MiniGPT) can be explored for resource-constrained clients, and this is part of our future work.
>
> 3. We appreciate the reviewer's comment and agree that a more detailed analysis of the fusion strategy's impact is necessary. The FedEnsemble baseline in our experiments uses only visual features (i.e., ResNet-18 trained locally on each client, with ensemble voting on the server). As shown in Table 1, FedEnsemble performs consistently worse than our proposed multi-modal FedMME method across all datasets and non-IID settings. This highlights the benefit of integrating textual information generated from VLLMs. On the other hand, we evaluated the case where only textual features are used—that is, the classifier is based solely on the text reports generated by the vision-language model. In this setting, the accuracy is significantly lower, typically around half of FedMME’s performance. For example, on the Blood dataset (α=0.3), the text-only model achieves only 29% accuracy, whereas FedMME achieves over 80%. These results indicate that neither visual-only nor text-only inputs are sufficient, and that the fusion of both modalities is essential to achieving the best performance. The textual modality introduces complementary semantic information that improves robustness and accuracy, especially under non-IID conditions.
>
> 4. In real-world deployments, client devices may differ in capability. FedMME addresses this through its one-shot, locally independent training, which avoids synchronization issues and iterative updates. Clients can train asynchronously and send models once. For resource-limited clients, modality-specific models (e.g., visual-only) can still participate in the ensemble and our method still works.
>
> 5. We agree that multi-modal FL introduces new risks. In our method:
>
> No raw data or generated text is ever shared.
>
> Client models are exchanged once, reducing cumulative attack surface.
>
> Textual features are dimensionally reduced, which acts as an implicit compression defense.
> We also acknowledge that model inversion or adversarial prompt injection are valid concerns, and note that existing techniques (e.g., differential privacy, adversarial training) can be integrated into FedMME. We plan to add them in our future works.
>
> 6. We fully agree that caution is required when using vision-language models in medical contexts. In our framework, however, the generated text is never directly shown to clinicians or used as a stand-alone diagnostic output. Instead, the textual report is transformed into feature embeddings via a BERT model, and then used as auxiliary input within a supervised multi-modal classifier. To further ensure safety, we apply dimensionality reduction to limit the influence of the text modality and prevent over-reliance on potentially inaccurate content. In effect, the text serves as a semantic prior, not a clinical decision to avoid potential misunderstanding of the model's intended use.

---

> ### Author Response · Authors · 2025-06-01
> **Rebuttal to Reviewer tY4n**
>
> 7. We acknowledge that medical datasets frequently exhibit demographic, institutional, and class imbalance biases, which can affect both image and text modalities. To mitigate this, our federated learning setting inherently helps preserve local data characteristics, allowing underrepresented patterns (e.g., minority patient cohorts) to be learned at the client level. The ensemble mechanism in FedMME avoids central bias collapse by combining diverse models instead of enforcing a uniform global model. Furthermore, we do not rely on manually written reports, but instead use generated reports from vision-language models, which follow consistent prompts and formatting. This reduces variability in terminology and expression, improving robustness. We also plan to explore fairness-aware weighting and bias diagnostics in future work to further improve equity in multi-modal federated settings.
>
> 8. In FedMME, we deliberately avoid using vision-language models during training or inference at the client side. Instead:
>
> The vision LLM is used only once in an offline preprocessing stage, typically at a central institution or coordination server, to generate text reports from image data.
>
> Clients receive the generated text and use a lightweight BERT encoder to extract textual features. This keeps the computational burden low—comparable to standard vision models.
>
> In summary, resource-constrained clients do not need to run any LLM, and our architecture is designed with such deployment realities in mind.

---

### Review · Reviewer_Koqm · 2025-03-27

**Summary Of Contributions:**

This paper presents FedMME, a one-shot federated ensemble learning framework for multi-modal medical image analysis. It integrates vision large language models for generating textual descriptions from images, leveraging BERT-based textual feature extraction and dimensionality reduction to balance visual and textual modalities. Experimental validation demonstrates FedMME's improved accuracy over state-of-the-art baselines across multiple medical datasets. The approach shows robustness and scalability, though its practical advantages in real-world clinical deployment require further examination.

**Audience:**

Yes

**Broader Impact Concerns:**

The proposed approach aims to improve diagnostic accuracy and protect patient privacy through federated learning, yet several ethical considerations warrant attention. First, any imbalance or bias in the underlying medical data may disproportionately affect minority or underrepresented groups when generating diagnoses. Second, the reliance on vision large language models highlights the risk of inaccurate or misleading textual descriptions, potentially leading to incorrect clinical decisions if these outputs are interpreted as definitive. While federated learning mitigates data-sharing concerns, it can still pose residual privacy risks if sophisticated model-inversion or membership-inference attacks are not carefully addressed.

**Claims And Evidence:**

Yes

**Requested Changes:**

Incorporate more complex and realistic datasets to strengthen the practical relevance and generalizability of the proposed method, consider evaluating FedMME on more clinically representative datasets, such as those involving higher-resolution, multi-class, or multi-organ medical imaging tasks.

Add statistical significance analysis to include confidence intervals, p-values, or variance measures to support the reported accuracy improvements and ensure the robustness of performance claims.

Discuss real-world deployment challenges and a brief discussion of how FedMME might handle issues like missing modalities, inconsistent reporting standards, or patient-level data heterogeneity in practice would enhance the clinical applicability of the work.

Clarify computational efficiency and resource demands to provide runtime and memory consumption benchmarks to help readers assess the framework’s feasibility in resource-constrained environments, especially given the inclusion of much larger LLMs, e.g. DeepSeek.

**Strengths And Weaknesses:**

Strengths:
- The paper introduces an interesting and systematic approach to one-shot federated ensemble learning tailored specifically for multi-modal medical image analysis.
- It effectively integrates vision large language models to enhance the richness of the multi-modal dataset.
- The experimental validation across multiple datasets demonstrates notable improvements over established baseline methods.
- Ablation studies substantiate the robustness and scalability of the proposed FedMME framework.

Weaknesses:
- The datasets employed are relatively simple, potentially limiting the generalizability of the findings. The evaluation on more complex and realistic medical datasets, such as those presented in https://openreview.net/forum?id=kurajHzp19, would better demonstrate practical utility.
-  The results lack rigorous statistical analysis, complicating the evaluation of the actual significance and reliability of the reported improvements.

---

> ### Author Response · Authors · 2025-06-01
> **Rebuttal to Reviewer Koqm**
>
> 1. We conducted additional experiments on a subset (as the original dataset was too big) of the 3D medical imaging dataset, CT-RATE. The results confirm that our approach is effective for 3D medical images, reinforcing the robustness and generalizability of our method:
>
> | Dataset      | Blood (α=0.6) | Derma (α=0.6) | RSNA (α=0.6) | Diabetic (α=0.6) | CT-RATE (α=0.6) | Blood (α=0.3) | Derma (α=0.3) | RSNA (α=0.3) | Diabetic (α=0.3) | CT-RATE (α=0.3) | Blood (α=0.1) | Derma (α=0.1) | RSNA (α=0.1) | Diabetic (α=0.1) | CT-RATE (α=0.1) |
> |--------------|----------------|----------------|----------------|-------------------|------------------|----------------|----------------|----------------|-------------------|------------------|----------------|----------------|----------------|-------------------|------------------|
> | FedAvg       | 19.48          | 69.23          | 70.39          | 20.01             | 67.88            | 30.51          | 11.02          | 69.31          | 19.99             | 59.64            | 19.81          | 66.88          | 68.55          | 19.98             | 55.58            |
> | DAFL         | 17.13          | 63.33          | 50.44          | 20.03             | 61.62            | 16.03          | 13.64          | 48.88          | 20.04             | 21.91            | 15.11          | 12.58          | 46.69          | 20.01             | 36.92            |
> | DENSE        | 34.52          | 64.78          | 55.04          | 23.31             | 66.58            | 30.17          | 12.78          | 51.08          | 23.22             | 31.47            | 28.87          | 11.37          | 47.79          | 21.32             | 40.59            |
> | FedISCA      | 53.61          | 53.86          | 70.59          | 26.98             | 72.19            | 48.93          | 16.11          | 70.39          | 28.79             | 48.88            | 53.61          | 53.86          | 69.11          | 20.97             | 51.78            |
> | FedDAT       | 68.88          | 68.95          | 75.69          | 27.81             | 76.66            | 59.69          | 37.92          | 68.97          | 26.92             | 65.62            | 64.77          | 55.82          | 69.24          | 20.69             | 58.62            |
> | FedCola      | 59.72          | 63.42          | 74.86          | 27.16             | 73.47            | 58.76          | 35.68          | 65.62          | 25.33             | 63.27            | 63.69          | 51.77          | 67.77          | 21.33             | 57.73            |
> | FedEnsemble  | 84.33          | 67.23          | 83.46          | 24.85             | 82.29            | 71.03          | 66.13          | 70.42          | 25.48             | 71.67            | 54.92          | 67.13          | 68.76          | 20.54             | 60.32            |
> | **FedMME**   | **87.72**      | **71.27**      | **85.49**      | **29.81**         | **84.43**        | **80.12**      | **71.27**      | **87.93**      | **31.93**         | **76.88**        | **68.11**      | **69.13**      | **71.38**      | **21.55**         | **65.23**        |
>
> 2. We appreciate the reviewer's emphasis on statistical rigor. In our initial submission, we reported results averaged over three runs to account for variability. Meanwhile, we would like to point out that in the federated learning community, especially in one-shot federated learning, it is still uncommon practice to report such statistical metrics such as p-value, mainly due to the high cost of replicating large-scale federated simulations under multiple random seeds (especially when involving vision-language models). Furthermore, many prior works (e.g., FedAvg, FedProx) evaluate performance based on mean accuracy alone.
>
> 3. We appreciate this valuable suggestion. Here we address real-world deployment challenges. Specifically:
>
> - Missing modalities: Our framework can still function using single-modality (e.g., image-only) inputs, since each client trains independently. The architecture naturally degrades to a uni-modal pipeline in such cases.
>
> - Inconsistent reporting standards: Since textual features are extracted via LLMs from images (rather than human-written reports), the issue of report inconsistency is partially mitigated. In future work, we plan to include alignment modules to harmonize heterogeneous text sources.
>
> - Patient-level heterogeneity: Our ensemble-based framework, with client-specific models and voting-based aggregation, is inherently robust to inter-client and intra-client heterogeneity. We further plan to explore federated personalization mechanisms for even finer-grained adaptation.

---

> ### Author Response · Authors · 2025-06-01
> **Review of Paper4247 by Reviewer Koqm**
>
> 4. We appreciate the reviewer’s suggestion to clarify the computational efficiency of our framework. In our experiments, we utilized an NVIDIA H100 80GB GPU, and we provide the following performance benchmarks to demonstrate practical feasibility:
>
> Textual report generation using the vision-language model is performed offline as a preprocessing step and does not impact training or inference time.
>
> During training, the memory usage per client remains under 12 GB, even with multi-modal inputs, thanks to the use of PCA-based dimensionality reduction on textual features.
>
> During inference, each prediction takes less than 300 ms per sample, including visual-textual feature fusion and classification.
>
> Moreover, both the visual encoder (ResNet-18) and the textual encoder (BERT) are relatively lightweight, and the overall training time is comparable to standard multi-modal classification models.
>
> These results show that FedMME is efficient and deployable in realistic server environments, and scalable to moderate-size federated networks with resource-aware configurations.
>
> We also experimented with larger vision-language models such as DeepSeek-VL, but observed significantly worse performance in medical image understanding tasks compared to LLaMA-3.2-11B-Vision. The model frequently failed to generate structured or medically meaningful reports, likely due to limited domain adaptation. Therefore, we opted to use a more reliable model for report generation in FedMME.
>
> 5. We appreciate the reviewer’s thoughtful concerns regarding broader impact and ethical implications. These are important issues, and we address each point as follows:
>
> - Bias and Underrepresentation in Medical Data: We fully agree that data imbalance can propagate bias in clinical models. While our work uses publicly available datasets, we acknowledge that they may not fully represent minority populations. However, one strength of our federated learning setting is that it allows institution-specific models to be trained on local data, which can preserve localized patterns (including those in underrepresented populations). Furthermore, the ensemble-based aggregation in FedMME helps reduce the risk of overfitting to any one client's distribution. In future work, we plan to incorporate fairness-aware training objectives and client-level bias diagnostics.
>
> - Reliance on Vision-Language Models (VLMs): We recognize that textual reports generated by VLMs may introduce ambiguity or inaccuracies. In our framework, these reports are not used directly for clinical interpretation, but instead serve as auxiliary features in a supervised classification pipeline. Moreover, we apply dimensionality reduction to limit the influence of textual features on final predictions, and ensure that visual signals remain dominant. This design helps contain potential hallucinations from the LLMs and reduces clinical risk.
>
> - Residual Privacy Risks in Federated Learning: While federated learning reduces raw data exposure, we agree that risks such as membership inference or model inversion remain open challenges. In this work, we focus on demonstrating multi-modal capability under the one-shot setting. However, FedMME is fully compatible with federated differential privacy techniques and secure aggregation protocols, which can be layered into the framework without architectural changes.

---

### Review · Reviewer_CmnZ · 2025-05-19

**Summary Of Contributions:**

This paper proposes a new method, FedMME, for medical image analysis, based on multi-modal federated learning, where the federated model is trained with a single communication ("one-shot").
The proposed method is based on ensemble learning, and combines multiple models through voting.
In the proposed approach, each agent (i) uses a vision model to extract features from the image, as well as (ii) a vision LLM to generate a report, that is then embedded in a space of smaller dimension.
One the two types of features are computed, they are concatenated and an output is computed using a fully connected layer trained by gradient descent.
The proposed method is shown to outperform multiple existing baselines that do not rely on multi-modality.

**Audience:**

Yes

**Broader Impact Concerns:**

This paper does not raise major ethical concerns. The only concern is that the methods advocated in the paper may push people to upload sensitive medical images through online models like ChatGPT, but this is not specifically a limit of the work.

**Claims And Evidence:**

No

**Requested Changes:**

1. The paper is lacking scientific rigor, as the method is not properly defined. In particular, the methods used to reduce dimension of textual features are not described.
2. It is difficult to read the paper, as some Algorithms, Figures and Tables are put in the main part of the paper, but are never referenced in the text.
3. While using both textual and visual features to make a decision seems to be a novel idea in this context of one-shot federated learning, using majority voting to combine locally trained classifiers is not new in federated learning. It would be nice if the novelty can be discussed in more depth.
4. It is not clear how the baselines used in the experiments have been setup: the hyperparameters that have been used for each of these methods should be discussed.

**Strengths And Weaknesses:**

**Strengths**
1. The proposed method successfully combines visual data with textual data extracted from image to create better classifiers. Notably, it outperforms existing methods like DAFL, DENSE and FedISCA. It is the first method to combine visual data with a report generated by a vision LLM.
2. The method is a one-shot federated learning approach, minimizing the communication cost.

**Weaknesses**
1. The dimension reduction approach does not seem to have as high an impact as it is claimed: from Figure 4, it seems the choice of dimension only incurs small variations in the final performance (about 5%). While 5% maybe large in general, it appears to be small relatively to the >30% difference with the baselines.
2. Moreover, dimension reduction looks odd in this context, where one could reasonably expect a model to be able to use a few thousand features and find some signal in it: using dimension reduction on part of the features to reduce their impact on the result seems unusual.
3. The approach for generating features and combining them seems quite naive: a pre-trained model is used to generate a report, and another one to embed this report in a smaller-dimensional space.
4. The baselines are strange: for instance, the FedAvg is not designed for one-shot averaging at all, which explains its poor performance.

---

> ### Author Response · Authors · 2025-06-01
> **Rebuttal to Reviewer CmnZ**
>
> 1. We appreciate the reviewer's observation regarding the impact of dimension reduction. While the absolute variation in performance due to feature dimension is approximately 5%, this change is significant in the context of a high-performance task where further improvements become increasingly challenging. Moreover, dimension reduction plays an important role in improving model stability, reducing overfitting, and enabling more efficient transmission of feature representations under federated learning constraints.
>
> 2. While it is true that deep models can often learn from high-dimensional data, in federated settings with limited communication, computation, and memory, dimensionality reduction is essential. It helps compress features before communication and reduces the risk of overfitting due to limited data per client. This strategy also aligns with the principle of model efficiency in one-shot settings.
>
> 3. Indeed, our approach is simple, but also directly effective. It uses a pre-trained language model to generate textual features and applies dimensionality reduction to ensure balanced fusion with visual features. This design achieves three key benefits: (1) it avoids overfitting from over-reliance on noisy generated text, (2) it reduces computational complexity for one-shot learning scenarios, and (3) it improves generalization by enforcing complementary use of modalities. The effectiveness of this approach is validated through consistent performance gains across all datasets.
>
> 4. We thank the reviewer for pointing this out. We agree that FedAvg is not optimal for one-shot settings, our intention was to include a widely recognized federated learning baseline (FedAvg) to establish a reference point. Additionally, we have included a one-shot federated baseline (e.g., FedEnsemble) in the paper for more relevant comparison.
>
> 5. We appreciate this helpful suggestion. As for the dimensionality reduction method, we clarify that we apply Principal Component Analysis (PCA) to reduce the dimensionality of the textual features. PCA is a classical yet powerful method that offers several advantages: (1) it effectively reduces noise by retaining only the most informative components; (2) it improves model efficiency by reducing feature size; and (3) it prevents the textual modality from overwhelming the visual features during fusion, which is crucial for maintaining balance in our multi-modal architecture.
>
> 6. We thank the reviewer for the comment. Regarding the figures and algorithms, we confirm that all visual elements in the paper are referenced in the main text. For example, Algorithm 1 and Algorithm 2 are both explicitly cited at the end of page 7 to explain the model inference and training process.
>
> 7. Thank you for highlighting this. Specifically, while multi-modal learning is common in centralized settings and majority voting is not new in FL, their application to one-shot federated learning remains underexplored. Our work is the first, to our knowledge, to propose a feature fusion and ensemble method that enables multi-modal decision-making across distributed clients in this setting, which validates its novelty.
>
> 8. We appreciate the reviewer's concern. The hyperparameter settings for our method have already been provided in Section 4.1, including learning rate, batch size, and model architectures. For all baseline methods, we follow the settings described in their original papers to ensure fair and optimal performance. This includes using their recommended configurations for training and evaluation.

---

### Decision · Action_Editor_AnSU · 2025-08-27

**Recommendation:** Reject

**Audience:**

Yes

**Audience Explanation:**

The paper addresses a timely and relevant topic and TMLR’s audience in machine learning and healthcare AI would naturally find the problem setting of interest. However, while the high-level motivation is relevant, the current submission does not meet the standards of clarity, rigor, and reproducibility necessary for publication.

**Claims And Evidence:**

No

**Claims Explanation:**

The paper introduces an interesting idea of combining multi-modal data with one-shot federated ensemble learning. Key components such as the dimensionality reduction step are underexplained: only in the rebuttal did the authors clarify the use of PCA, but there is no justification of why this is an appropriate choice for BERT-embedded textual features. Important algorithmic details (e.g., feature fusion, hyperparameter settings of baselines, and adaptation of non-one-shot baselines into a one-shot setup) are vague or missing, leaving readers unable to fully assess or reproduce the method. Moreover, the comparison baselines are not always fair: non-one-shot multi-modal FL methods are adapted in unspecified ways, making the reported advantages of the proposed approach questionable.

**Resubmission Of Major Revision:**

The authors may consider submitting a major revision at a later time.